# Intensity-Dependent Effects of a Six-Week Balance Exercise Program in Elderly Women

**DOI:** 10.3390/ijerph15112564

**Published:** 2018-11-16

**Authors:** Zbigniew Borysiuk, Paweł Pakosz, Mariusz Konieczny, Krzysztof Kręcisz

**Affiliations:** Faculty of Physical Education and Physiotherapy, Opole University of Technology, Prószkowska 76 Street, 45-758 Opole, Poland; z.borysiuk@po.opole.pl (Z.B.); m.konieczny@po.opole.pl (M.K.); k.krecisz@po.opole.pl (K.K.)

**Keywords:** limits of stability, coactivation, EMG signal, falls, aging

## Abstract

The objective of this study was to gain a better understanding of the mechanisms underlying falls in the elderly. The results were based on a group of 28 women in a control group (CON) and 16 women in an experimental group (EXP), aged 60–70. Participants took part in the six-weeks Elderly Recreation Movement Program (ERMP) with the only difference that the EXP group practiced twice as often as the CON group. The measurement of variations in the index called limits of stability (LOS) was performed by application of Kistler force plate and the coactivation index (CI) was registered by means of sEMG. The results demonstrate the existence of statistically significant differences in terms of the principal outcome of the exercise time in the measurements of LOS (F(1.42) = 10.0, *p* = 0.003), and CI (F(1.42) = 10.5, *p* = 0.002). The effect of the program was associated with an increase the level of the maximum LOS, and a decrease of the CI level, especially in the experimental group. Hence, the implementation of an innovative ERMP exercise program results in the improvement of the physical capabilities of senior subjects.

## 1. Introduction

The aging process is associated with losses in the mechanisms of neural transmission, which results in a decrease of compensatory behaviors applied in the maintenance of postural stability, thus leading to a greater expenditure in terms of cognitive resources [1]. The limitation of the deficiencies expressed in motor abilities converts into lower risk of falls, which otherwise potentially result in more or less serious injuries that can deteriorate body functions. The subject of falls forms a very important issue in the context of the aging society, as the fear of falling is known to increase with age [2].

The aging process also brings about a decrease in the control over the body and when they are faced with a need to deal with a task with a greater complexity, seniors need to overcome problems in the integration of information from multiple sensory systems [3]. The older age also brings about an increasing risk of falls with it, which socially pose a serious health problem and form one of the main reasons for injuries, disabilities and even deaths. It is estimated that one third of older people living in the modern communities experience at least a single fall a year [4]. In addition, studies on fall profiles show that more than half of the causes of falls are related to internal factors that are associated with a loss of the postural control [5].

Postural control is understood as a complex relation between the sensory and motor systems [1]. This involves, among others, an adequate perception of external stimuli, reaction to changes of the body orientation in space, as well as maintenance of the center of pressure exerted by the body. The elderly face deficiencies in the coordination of movements, and the movement patterns become slower and less fluid [6]. The difficulties associated with maintaining the balance are also attributed to the weaker control expressed by the displacement of the center of pressure [7].

The issues related to balance can be described in terms of the parameters such as center of pressure (COP) and limits of stability (LOS). COP is defined as point of application of the force exerted by feet on the ground, [8] and (LOS) can be described in terms of the maximum distance over which a person can transfer their center of pressure by leaning the body in a specific direction on condition that this process does not result in the loss of balance. In addition to determining the parameters related to postural stability, the activity of the lower leg muscles is also important, as it directly affects the ability to return to the vertical position after balance is lost [9,10]. The activity of the lower leg muscles is assessed by muscle co-activation (interaction of agonist and antagonist muscles), which varies with age and results in an increased synergistic and antagonistic muscle behavior [11].

The studies involving the displacement of COP in various conditions in senior-aged subjects [4,12], as well as determination of the most adequate methods to be applied to control the reliability of the control of limits of stability (LOS) [13] can contribute to the development of preventive therapy programs for seniors designed so as prevent falls in them [14]. In addition, research was also carried out to determine the dependencies of both parameters and their variations accompanying the aging processes [15,16,17].

The objective adopted in the research reported in this paper involved the analysis of the influence of a six-week exercise program on improving stability and change of muscle co-activation in senior-aged women. The innovative nature of the present program involved the application of a balance training combined with the ERMP program (Elderly Recreation Movement Program) developed so as to provide seniors with a practice of postural balance exercise in the AP plane on an unstable ground. A hypothesis was adopted that the doubling of the intensity of exercises focusing on balance control will lead to the improvement in terms of the ability of body displacement in the examined subjects, and the co-activation of muscles responsible for control of the ankle will decrease.

## 2. Patients and Methods

The research involved a group of 60 women aged 60–70 years, belonging to an active senior club actively operating in the provincial town of Opole. Applications for research were preceded by a recruitment program conducted in the environment of seniors in Opole (promotional and educational campaign was carried out for this purpose), under the assumption that the participants will include seniors who have not been involved in regular physical activity. The condition for participation in the program was also associated with the lack of health contraindications and the voluntary participation of the senior women. The details of the inclusion and exclusion criteria are summarized below.

Criteria for patient participation in the study:Women aged 60–70 years;Ability to understand instructions and active participation in the tasks (Mini-Mental State Examination < 23 p.);Capacity to actively participate in physical exercise (Elderly Recreation Movement Program—ERMP);Ability to move independently and lack of injuries to lower limbs;Lack of medical contraindications to participate in physical exercise with moderate intensity;Agreement to participate in the study.

The criteria adopted as the basis for the rejection of an applicant included:Aphasia;Significant loss of sight or hearing, which makes it impossible to assess cognitive functions;Withdrawal from the study.

The subjects were divided into two equal groups: control (CON) and experimental (EXP) ones. The decision regarding assigning women to specific groups was random and researchers did not intervene with the decision when the subjects objected to participation in a given group. The fact of assigning subjects to a particular group was not affected in any way by the results gained by them in the initial test. For the reasons independent of the course of the experiment, 28 CON and 16 EXP seniors completed the study (i.e., in the cases when subjects resigned for personal purposes). The number of subjects in the experimental group decreased to 16 since only the results of the women who were able to achieve 100% attendance in the exercise program were taken into account. A summary of information regarding the age, height and weight of the subjects is included in Table 1. The subjects selected for the study were also asked not to participate in other forms of physical activity for the duration of the program. The duration of the program equal to 6 weeks was determined on the basis of literature analysis describing the effectiveness of interventions in the elderly subjects and by the schedule of seniors’ environment in Opole.

Subjects in both groups took part in the rehabilitation exercises with the only difference being that the experimental one took part in the exercises twice weekly for 60 minutes, whereas the control group only once per week with the same duration. Every time, the exercise program was based on a similar schedule comprising a warm-up, core part and a final part focusing on muscular relaxation and relieve of tension. Classes were organized in a room suitably adapted to this type of exercise, equipped with in mirrors, handrails, ladders and other suitable equipment. The program also included equivalent classes on stable and unstable surface as described in Table 2. The participants of the study signed an informed consent. The goal of the study was approved by the Bioethics Committee of the Chamber of Physicians (Resolution No. 237 of 13 December 2016) with accordance to the guidelines described in the Helsinki Declaration involving humans.

The ground reaction forces was recorded with the force platform and the bioelectric activity of the right and left tibialis anterior (TA) and soleus muscles (SOL) was registered and processed by NORAXON sensors (DTS type, Noraxon, Scottsdale, AZ, USA) with the procedure followed SENIAM methodology.

The subjects in both groups were measured prior to and after the exercise program. The maximum voluntary contraction (MVC) of the examined muscles were performed over a 5 second voluntary contraction and was used as a reference value. The outcome variables were measured in maximal voluntary forward lean with eyes open (LOS) maintained for 10s. The instantaneous center of foot pressure (COP) were calculated from the components of forces of the registered plate response were analyzed in anterior-posterior planes (AP). The registrations of COP were performed by a force plate (Kistler type 9286AA, Winterthur, Switzerland), with a sampling frequency of 100Hz, test duration: 30s. The electromyographic study applied 16 channel sEMG signals (DTS type, Noraxon, Scottsdale, AZ, USA) recorded with the 16 bit resolution and a sampling frequency of 1500 Hz.

The higher values of LOS derived from difference of mean 10s before leaning and maintained in maximal forward lean values indicate a more effective postural control [13]. The sEMG signals were smoothened by estimation of the root mean square (RMS) that was derived in the time window of 300 ms. The reference value of the MVC was calculated in a time window equal to 1000 ms for which the mean value of the sEMG signal was the highest.

The coactivation (CI) of the muscles was calculated using the Falconer and Winter method. The CI was computed separately for the left (CI L) and right (CI R) legs.

The indices of the COP and sEMG signals were subjected to the Shapiro-Wilk normality test. The distributions of the analyzed variables did not deviate from normality. Thus the parameters of signals were subjected to ANOVA (analysis of variance) with 0.05 significance thresholds. The factors included: time (before, after) for the case of analysis of LOS and additionally side (left vs right) for the case of the coactivation index.

## 3. Results

The results of the analysis of the LOS demonstrate statistically significant differences before and after the series of exercises in both groups (principal outcome TIME: F(1.42) = 6.337, *p* = 0.002). The interaction of the variables TIMExGROUP did not indicate statistically significant differences. Table 3 contains the mean values and standard deviations of LOS values for both groups. The data in Figure 1, on the other hand, demonstrates an increase in the mean LOS values following the exercise program with regard to both subjects’ eyes open and closed.

The analysis of the CI parameter does not demonstrate significant differences measured in terms of the principal outcome represented by the duration of the exercise (F(1.42) = 3.501, *p* = 0.069). Both groups demonstrate a decrease in the level of muscle coactivation following the exercise program. The remaining effects do not show statistically significant results. Table 4 contains the mean values and standard deviations of the CI parameter for both groups. The results in Figure 2 demonstrate that the mean values of CI following the exercise program have decreased. We can note that the subjects in the experimental group show greater falls; however, they do not assume statistically significant values in comparison to the control group.

## 4. Discussion

The research shows the effectiveness of the adapted rehabilitation program in the seniors, whose aim was to contribute to preventing falls. The research conducted earlier on, in which LOS was also applied as an experimental tool demonstrates that it forms a reliable indicator for the analysis of the range of the forward body displacement, which is directly related to the chance of falls [13,18]. Correlations between the results of such tests as well as diagnostics performed into seniors who have had previous fall instances are described in a study by Lázaro et al. [4]. The authors of this study also believe that this type of research forms the key to taking adequate measures so as to prevent further falls, and also leads to a decrease of their physical, psychological and social consequences.

The results reported in this paper demonstrate a significant improvement in terms of the measured increase of the range of a forward displacement (tilt), which may directly contribute to prevention of falls and greater reliability in terms of the movement patterns performed by seniors in general. A statistically significant improvement in LOS was also recorded by the subjects involved in the program designed by Bulat et al. [19] following a six-week rehabilitation program; however, that study involved a smaller duration of exercises per week compared to the present program.

The above aspect of the usefulness of LOS measurements and the effectiveness of interventions in the field of stabilography was also recorded by Narita et al. [20], who applied a similar course of exercises as in the present program. The authors also put emphasis on exercises that seniors are capable of practicing at home so as to achieve an improvement of LOS following an exercise program lasting 12 weeks.

The next important aspect of maintaining a vertical posture is associated with the adequate level of co-activation of the lower leg muscles. The results of the present study led to improvement of the subjects’ sense of balance and the change in calf muscle activation indicates a transition from the less effective and energy-consuming strategies involving maintenance of balance through simultaneous muscle activation of antagonist muscles in the ankle to a more energy-efficient strategy of the alternate activation of muscles. The studies study by Pereira and Goncalves [21] as well as by Nagai et al. [10] include a recommendation that the rehabilitation and intervention programs performed with the purposes of reducing the risk of falls in the elderly should include exercises aimed at reducing co-activation. This offers a beneficial aspect since in the opinion of another author [22] excessive muscular contractions may potentially lead to a higher risk of postural instability. We should also note that greater muscle contractions often involve the risk of fatigue and a decrease in the efficiency of a projected task. Hence, a high priority in seniors may be associated with the need to compensate for the deterioration associated with postural control, since increased muscle co-activation is associated with weaker motor control [23].

There are also other reports [24,25], in which the authors suggest that the intervention program undertaken with the purpose of preventing falls should be aimed at strength training of the muscles of the ankle. Such training is designed to increase the maximum muscle strength of the subjects, which is aimed at reducing the relative muscle input needed to maintain balance in the upright position. As a result of the designed six-week exercise program, Penzer and colleagues [26] demonstrated that both strength as well as balance training affect the increase of muscle strength and improvement of COP parameters to a similar degree.

The results of the present intervention program aimed at preventing falls demonstrate that the participants’ confidence in their ability to displace the body forward has increased as a result of the exercise practice, and this aspect is known to play an important role in everyday life. Our activities have also confirmed the assumptions that should be met by the exercise program, as recommended by Sherrington et al. [27]. Gillespie at al. [28] believe that organized and individual exercises and interventions programs followed at home reduce the risk of falls, and multidisciplinary intervention programs decrease the incidence and risk of falls. Exercise programs that can be carried out at home are of particular importance because more than half of the falls at home conditions are directly life-threatening [29]. In addition, physically active seniors are able to apply additional brain resources to improve the performance of various activities at home [30]. The aerobic aspect implemented in the exercises improves the performance of the central nervous system, increases brain plasticity and improves cognitive functioning in the elderly [31]. For these reasons, it is advisable to convince elderly people to take part in this kind of exercises, so that they can feel more confident when performing various everyday life activities. If a physical exercise program is performed regularly and under the supervision of qualified personnel, it can only bring benefits. In addition, a higher level of physical activity affects the high self-esteem of the general quality of life of older people [32,33,34].

Throughout the duration of the six-week program, the experimental group performed exercise twice as often as the control group. This forms the reason why one of the objectives of this study involves making sure how the duration of the exercise affected the changes in analyzed parameters. On the basis of a study carried out by Sherrington at al. [35], the minimum effort that can potentially results in reducing the risk of falls is 50 hours of exercise; yet, in our program the total duration of the exercise was shorter. The results of programs with shorter duration are inconclusive, since in some of them the risk of injury did not change significantly [36,37], whereas in others the risk dropped significantly [26,38]. Following the exercise program designed for this study, the results in both groups were improved both in terms of LOS parameters as well as in terms of muscle coactivation. Whereas the inter-group comparison does not indicate a change in terms of the LOS, the co-activation level was clearly lower in the group that exercised more often. This proves the greater susceptibility of the muscle coactivation to exercise compared to the effects measured in terms of LOS.

The present approach involves several aspects that affect the limitations of the results. First of all, we did not measure sEMG of other lower limb muscles because the tasks focused on the ankle joint. Further research would be needed to extend the range of the analyzed joints and muscles. Another, but slightly controversial, limitation could be associated with the static conditions in which the research was conducted; as such conditions represent the actual conditions of everyday life of the elderly only to a limited degree. However, the present study demonstrated that if muscle activation increases so as to match the need to maintain body balance, this increase should be noticeable both in static and dynamic conditions and should be related to the level of body’s ability to keep control of the posture [39]. During everyday activities, both static and dynamic postural control needs to be successfully implemented. We can also raise doubts regarding the selection of the duration of the experiment, since various research by other authors has shown [40] that that even a significant increase in the duration of the exercise can reduce the risk of falls by up to 50%. However, in the conditions when longer experiment are followed, an increased number of side factors cannot be excluded, because in such cases it is more difficult to account for other activities undertaken by the examined persons that are not projected within the analyzed program. A significant variable affecting the outcomes of the intervention is also associated with the form of the exercises that have been applied, as better results are reported following, e.g. Tai Chi exercises, than after conventional stabilography exercises in other works dealing with this topic [22,41]. Nevertheless, the fact that the chances of the elderly to participate in sophisticated forms of activity such as Tai Chi are much smaller than in conventional exercises, as the number of the coaches capable of developing programs similar to the present exercise is greater and the accessibility to such activities is much more common.

## 5. Conclusions

The innovative ERMP exercise program forms an effective tool in improving balance in the elderly subjects. The outcome of the present program was represented by an increase of the level of maximum LOS, and a decrease in the CI level particularly in the subjects in the experimental group.

## Figures and Tables

**Figure 1 ijerph-15-02564-f001:**
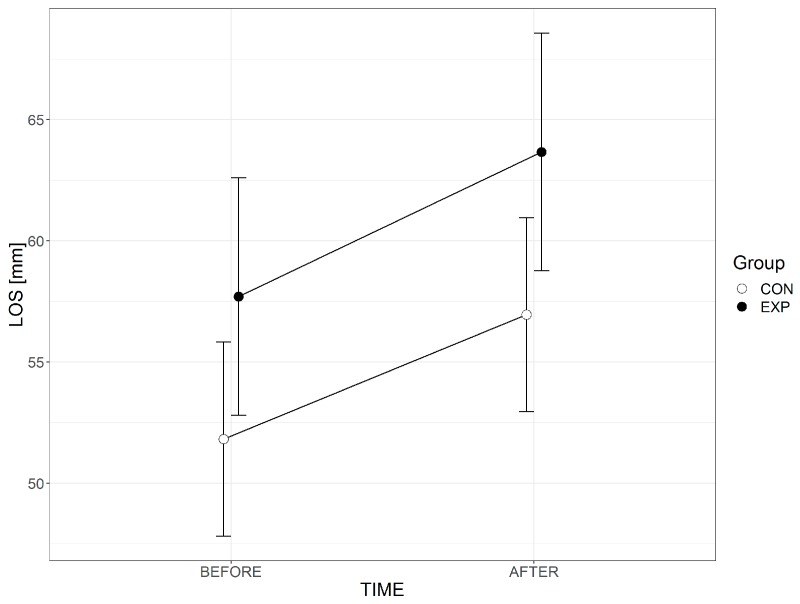
Graphical representation of the results of LOS.

**Figure 2 ijerph-15-02564-f002:**
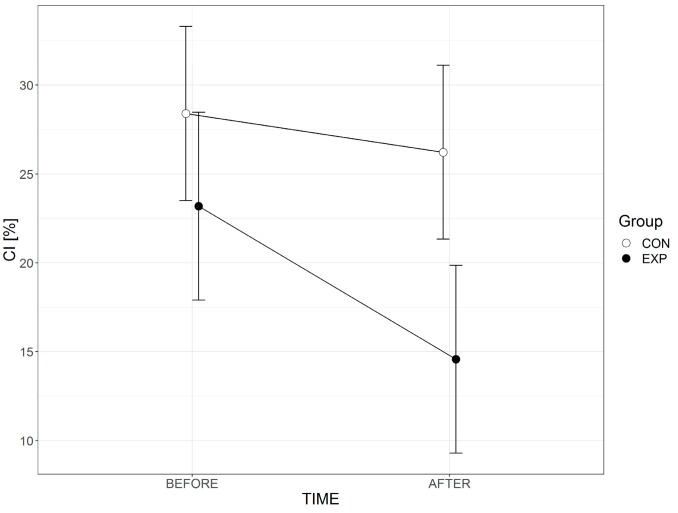
Graphical representation of interaction TIMExGROUP, whiskers indicate 95% confidence intervals as the combination of right and left legs.

**Table 1 ijerph-15-02564-t001:** Characteristics of examined group (mean ± standard deviations).

Group	Number of Participants	Age	Height	Weight
		(Years ± s.d.)	(cm ± s.d.)	(kg ± s.d.)
Control group (CON)	28	69.5 ± 4.6	161.9 ± 7.2	74.2 ± 11.4
Experimental group (EXP)	16	66.4 ± 5.9	159.0 ± 6.1	70.2 ± 13.9

**Table 2 ijerph-15-02564-t002:** Details of balance exercise program.

1) Standing with legs slightly apart, and slightly bent knees. The center of pressure is transferred forwards and back (toes to heels).
2) Standing with legs slightly apart with both arms extended to the side, alternating leg bends in the hip and knee joints up to 90°.
3) Standing with legs slightly apart and supported by the ladder Alternately, knees are lifted to reach to the bent elbow of the opposite arm.
4) Standing with legs slightly apart on a 10 cm foam pad, legs are alternately bent in the hip and knee joints up to 90°—with open eyes and next with closed eyes.
5) Standing with legs slightly apart on a 10 cm foam pad, legs are alternately bent in the hip and knee joints up to 90° (for 30 s)—with open eyes and then with closed eyes.
6) Exercise in pairs. A slight nudge is given to person holding the posture of an inverted pendulum. The task of the subject is to return to a vertical position.
7) A march forward with a 360° rotation in the indicated direction in response to a signal of the coach, the march is continued after the turn.
8) March in reverse directions (forwards and backwards).
9) March along a variable ground (mattress, bench).

**Table 3 ijerph-15-02564-t003:** Mean values and standard deviations of LOS.

Measure	Before LOS		After LOS	
	CON	EXP	CON	EXP
Mean	51.82	57.71	56.95	63.66
Std. Deviation	17.13	21.55	17.05	26.78

CON—control group, EXP—experimental group.

**Table 4 ijerph-15-02564-t004:** Mean values and standard deviations of CI parameter.

Measure	Before CI R		Before CI L		After CI R		After CI L	
	CON	EXP	CON	EXP	CON	EXP	CON	EXP
Mean	27.76	24.48	29.03	25.59	27.57	15.29	25.15	15.81
Std. Deviation	16.45	20.02	19.03	16.16	19.02	15.65	20.39	13.59

R—right leg, L—left leg, CON—control group, EXP—experimental group.

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
