# Peer review of "Intensity-Dependent Effects of a Six-Week Balance Exercise Program in Elderly Women"

_ijerph, 2018, doi:10.3390/ijerph15112564_

Reviewer 1 Report

Any measure directed to prevent falls in older people is worth to be considered. In the present work, the authors show the effects of a program based on physical exercises to improve stability and muscle co-activation.

I have some comments, doubts and questions related to the manuscript:

-          The abstract, besides results should also contain the objective, methods and conclusions of the work

-          Durations of the program, it is indicated in the last paragraph of the introductions, but it should be included in the methods section. In what basis was determined that the intervention should last 6 weeks?

-          Materials and methods, it would be better to name this section, patients and methods

-          Why did the authors, only included women in the study?

-          The number of patients included is very low, especially in the intervention group.

-          Criteria of inclusion and rejection should be following the explanation in the text, line 78 or referred as a table.

-          The discussion section is too long, it should be reduced and focused in the main findings of the work

Author Response

Point 1: The abstract, besides results should also contain the objective, methods and conclusions of the work

Response 1: Following the remark provided by the Reviewer, we have adapted the first sentence of the abstract so as to include a more accurate determination of the objective of the study: „The study objective was to gain a better understanding of the mechanisms underlying the falls in the elderly”. In our opinion, the objective, methods and conclusions are included in the body of the abstract, but the requirement imposed of the journal restricting this section to 200 words provides a limit to the broader description of the raised issues.

Point 2: Durations of the program, it is indicated in the last paragraph of the introductions, but it should be included in the methods section. In what basis was determined that the intervention should last 6 weeks?

Response 2: According to the Reviewer's suggestion, we have also included the details regarding the duration of the program in the Methods section. The six-week duration of the program was determined by the availability of subjects in the Active Senior’s association.

Point 3: Materials and methods, it would be better to name this section, patients and methods

Response 3: Based on the suggestion provided by the Reviewer, we have changed the title of this section to “Patients and Methods”

Point 4: Why did the authors, only included women in the study?

Response 4: The research only included women as the projected scope of study that was followed in the project envisaged the inclusion of as homogeneous group of subjects as it could be possible.

Point 5: The number of patients included is very low, especially in the intervention group.

Response 5: The small number of people in the examined group was associated with the limitations resulting from the course of the intervention program, as well as logistics and financial restraints imposed by the location of the study and with limited accessibility to a coach suitably trained to conduct the exercise program.

Point 6: Criteria of inclusion and rejection should be following the explanation in the text, line 78 or referred as a table.

Response 6: We have adapted the manuscript to follow the suggestion of the Reviewer.

Point 7: The discussion section is too long, it should be reduced and focused in the main findings of the work

Response 7: We have adapted the manuscript and deleted the paragraphs between lines 188-198 and 222-227.

Reviewer 2 Report

The main objective of this study was to evaluate effectiveness of a six-week balance exercise program in women seniors. Researchers implemented a set of an exercise program and tested this program in a cohort. Despite results showed a relative improvement in postural stability, the study has shortcomings which some were stated in discussion session. The main criticism is, from a health profession perspective, how this exercise program is effective in reducing risk of falls in seniors. Most of the balance exercise programs are effective in postural stability however the current study does not show any innovative difference. In other words, what part of this exercise program is considered in an innovation manner, comparison to existing exercise programs for fall prevention?

The manuscript is well-written, however I would encourage authors to incorporate following findings,

Page 4, line 122, Please justify why researchers decided to analysis only CoP in AP directions. Since researchers collected muscle coactivation on both sides, it may lead to ML component of CoP which is an important parameter for postural sway.

Page 5 Fig1. Please provide more information regarding the test. Despite both groups showed relative improvement in LOS, would it be possible to explain why both groups did not have similar LOS for the initial condition or Before?  What is the norm of LOS in this particular population? Please add more ticks for intervals on y-axis.

Page 6 fig1 (should be Fig2. Please fix), what are those groups’s CI? Right, left or combination of right-left?

Page 6 table 4. Standard deviation of CI parameter is too big, it is 100% for After CI right Exp group. Is there any experimental error or calculation error?

Page 7, line 178, Please use “may” instead of “can” since this is a pilot study despite there is an improvement in balance performance.

Author Response

Point 1: The main objective of this study was to evaluate effectiveness of a six-week balance exercise program in women seniors. Researchers implemented a set of an exercise program and tested this program in a cohort. Despite results showed a relative improvement in postural stability, the study has shortcomings which some were stated in discussion session. The main criticism is, from a health profession perspective, how this exercise program is effective in reducing risk of falls in seniors. Most of the balance exercise programs are effective in postural stability however the current study does not show any innovative difference. In other words, what part of this exercise program is considered in an innovation manner, comparison to existing exercise programs for fall prevention?

Response 1: We have modified an exercise program designed for seniors, one that is commonly applied in Poland and was developed by prof. Kozdroń (Borysiuk et al), and have introduced stabilographic exercises, which according to literature data can reduce the incidence of falls of the elderly. The innovation was associated with the combination of exercises aimed at balance training with the program developed by Prof. Kozdroń named ERMP (Elderly Recreation Movement Program) involving the generation of sways in the AP plane on an unstable ground.

Point 2: Page 4, line 122, Please justify why researchers decided to analysis only CoP in AP directions. Since researchers collected muscle coactivation on both sides, it may lead to ML component of CoP which is an important parameter for postural sway.

 Response 2: The selection of the AP plane results from the effectiveness and reliability of the LOS measurements reported in the literature (Juras et al., 2008)

Point 3: Page 5 Fig1. Please provide more information regarding the test. Despite both groups showed relative improvement in LOS, would it be possible to explain why both groups did not have similar LOS for the initial condition or Before?  What is the norm of LOS in this particular population? Please add more ticks for intervals on y-axis.

Response 3: We think that these tests are meaningless, since if the groups were allocated randomly we know the null hypothesis, which is true about the population, not the sample.  An additional factor that was found interesting was associated with teh GROUPxTIME interaction, which involved the analysis if subjects in the particular groups have improved their results. We are not familiar with any norms regarding LOS.

The charts were supplemented to include whiskers to account for the intervals on y-axis, as required by the Reviewer.

Point 4: Page 6 fig1 (should be Fig2. Please fix), what are those groups’s CI? Right, left or combination of right-left?

Response 4: Fig 2. deals with the combination of the right and left legs.

Point 5: Page 6 table 4. Standard deviation of CI parameter is too big, it is 100% for After CI right Exp group. Is there any experimental error or calculation error?

Response 5: We have checked the data again and the calculations are correct. The result is due to the differentiation in the group of the examined women and level of physical activity of the subjects.

Point 6: Page 7, line 178, Please use “may” instead of “can” since this is a pilot study despite there is an improvement in balance performance.

Response 6: We followed and changed the manuscript to follow the suggestions provided by the Reviewer.

Reviewer 3 Report

General comments

In their manuscript: “Effect of the application of an innovative program on prevention of falls in seniors” Borysiuk Z. et al. present data from 43 elderly female participants of a six-week balance exercise program with varying intensity. Please find below a detailed list of comments raised during review of the manuscript.

Detailed comments:

Introduction: l.65-70: The study objective and hypothesis do not agree with the study’s design (testing for differences in effect size related to doubled intensity in the experimental vs. control group).

Methods: was group allocation randomised? Because Table 1 suggest significant group differences with regard to age and BMI (please statistically test for group differences) between experimental and control group; and also Figure 1 and Figure 2 should depict identical starting points at baseline (given randomized group allocation).

Methods: Why did the authors choose to refrain from an intervention-free control group?

Results: the presented differences between experimental and control group simply reflect differences in intensity, but are interpreted as proof of effectiveness (l. 20: “…results in the improvement…” & l.282-283), although this conclusion would only be valid with an intervention-free control group.

The study assessed limits of stability (LOS) and the coactivation index (CI) as surrogate markers of balance, but not the incidence of falls related to the intervention. Thus, I would suggest to not over-interpret the present data and speculate about the effectiveness of the investigated intervention for the prevention of falls – this outcome was not assessed.

Title: suggest to revise to more precisely “Intensity-dependent effects of a six-week balance exercise program in elderly women”

Table 1 outlines 28 participants in the control group, but the abstract N = 27 – please clarify.

Author Response

Point 1: Introduction: l.65-70: The study objective and hypothesis do not agree with the study’s design (testing for differences in effect size related to doubled intensity in the experimental vs. control group).

Response 1: The objectives and hypotheses were adapted to follow the remark of the Reviewer regarding the new title of the paper. The relevant changes are now included in the final paragraph of the Introduction section.

Point 2: Methods: was group allocation randomised? Because Table 1 suggest significant group differences with regard to age and BMI (please statistically test for group differences) between experimental and control group; and also Figure 1 and Figure 2 should depict identical starting points at baseline (given randomized group allocation).

Response 2: We think that these tests are meaningless, because if the groups were allocated randomly we know the null hypothesis, which is about the population, not the sample, is true. An additional factor that was found interesting was associated with the GROUPxTIME interaction, which involved the analysis if subjects in the particular groups have improved their results.

Point 3: Methods: Why did the authors choose to refrain from an intervention-free control group?

Response 3: The lack of an involvement of an intervention-free control group  came as a result of the small number of subjects available for the study. This was associated with the limitations resulting from the course of the intervention program, as well as logistics and financial restraints imposed by the location and with limited accessibility to a coach trained to conduct the exercise program.

Point 4: Results: the presented differences between experimental and control group simply reflect differences in intensity, but are interpreted as proof of effectiveness (l. 20: “…results in the improvement…” & l.282-283), although this conclusion would only be valid with an intervention-free control group.

Response 4: The applied exercise program led to an improvement of the results related to stability in both groups, and this is an indisputable result that could be concluded from the LOS analysis. We agree with the Reviewer’s statement that the intensity did not significantly change the results between the group; however, we think that this is due to the variation in the level of physical capacities of the subjects. As a consequence of the implementation of an exercise program, the authors intended to verify the impact of the exercises and their intensity on improving stability.

Point 5: The study assessed limits of stability (LOS) and the coactivation index (CI) as surrogate markers of balance, but not the incidence of falls related to the intervention. Thus, I would suggest to not over-interpret the present data and speculate about the effectiveness of the investigated intervention for the prevention of falls – this outcome was not assessed.

Response 5: We agree with this remark and for this reason we deleted the statements regarding the impact of the program on the prevention of fall from the following sections in the paper: Abstract and Conclusions.

Point 6: Title: suggest to revise to more precisely “Intensity-dependent effects of a six-week balance exercise program in elderly women”

Response 6: The title was adapted in accordance with the suggestion of the Reviewer.

Point 7: Table 1 outlines 28 participants in the control group, but the abstract N = 27 – please clarify.

Response 7: The error in the abstract was corrected, i.e. now the number is correct: N=28.

Round  2

Reviewer 1 Report

The authors have improved the manuscript according to the suggestions given.

Reviewer 2 Report

Thank you very much for the edits, no further concerns, 

Reviewer 3 Report

All comments raised during review were addressed by the authors.